# Exploring Microstructure, Wear Resistance, and Electrochemical Properties of AlSi10Mg Alloy Fabricated Using Spark Plasma Sintering

**DOI:** 10.3390/ma16237394

**Published:** 2023-11-28

**Authors:** Guangfei Rong, Wenjie Xin, Minxu Zhou, Tengfei Ma, Xiaohong Wang, Xiaoying Jiang

**Affiliations:** Key Laboratory of Air-Driven Equipment Technology of Zhejiang Province, Quzhou University, Quzhou 324000, China; rongguangfei2023@163.com (G.R.); xinwenjie0709@163.com (W.X.);

**Keywords:** SPS, aluminum alloy, microstructural evolution, wear property, electrochemical property

## Abstract

Al-Si-Mg alloy has excellent casting performance due to its high silicon content, but the coarse eutectic silicon phase can lead to a decrease in its mechanical properties. Samples of AlSi10Mg alloy were prepared by using a spark plasma sintering method, and it was found that sintering temperature has a significant impact on the grain size, eutectic silicon size and wear and corrosion properties after heat treatment. At a sintering temperature of 525 °C, the alloy exhibits the best wear performance with an average friction coefficient of 0.29. This is attributed to the uniform precipitation of fine eutectic silicon phases, significantly improving wear resistance and establishing adhesive wear as the wear mechanism of AlSi10Mg alloy at room temperature. The electrochemical performance of AlSi10Mg sintered at 500 °C is the best, with I_corr_ and E_corr_ being 1.33 × 10^−6^ A·cm^−2^ and −0.57 V, respectively. This is attributed to the refinement of grain size and eutectic silicon size, as well as the appropriate Si volume fraction. Therefore, optimizing the sintering temperature can effectively improve the performance of AlSi10Mg alloy.

## 1. Introduction

Aluminum alloys are widely used in various industries such as aerospace, automotive, weaponry, electronic packaging, and chemical industries due to their low density, high specific strength, and excellent corrosion resistance [1,2,3,4,5]. The Al-Si-Mg system is particularly notable for its strength, plasticity, corrosion resistance, and good flowability, making it a common cast aluminum alloy. However, the coarse eutectic silicon phase directly degrades its mechanical properties, especially its toughness. Additionally, eutectic silicon tends to increase with increasing silicon content [6,7]. Therefore, it is necessary to add certain modification agents during the casting of Al-Si-Mg alloys to control the grain size and morphology of eutectic silicon [8,9,10].

According to Jia et al. [11], adding Sr and Y elements can improve the spheroidization and grain refinement of eutectic silicon. After adding 0.2% Y aluminum silicon alloy for heat treatment, the tensile strength and elongation increased by 14.3% and 8.3%, respectively, to 344 MPa and 6.3%. Yin et al. [12] reported that adding Sn can refine and modify eutectic silicon particles. Heat treatment is also used to improve mechanical properties and spheroidized Si particles [13]. However, the relationship between the eutectic silicon, electrochemical performance and grain size in Al-Si-Mg alloys is complex. In addition, for cast aluminum alloys, it is also necessary to consider the influence of secondary dendrites [14]. Kim et al. [15] studied the effect of ultrasonic melting treatment on the corrosion properties of A356 alloys. The effects on the corrosion properties were attributed to the uniform dispersion of refined Al grains, SDAS, and eutectic Si. Specifically, when the morphology and distribution of silicon particles are more uniform, corrosion resistance is improved. In addition, a more uniform and fine distribution of silicon particles is beneficial for improving the corrosion resistance of Al-Si alloys [16]. Therefore, controlling the size and morphology of eutectic silicon in Al-Si-Mg alloy is crucial for microstructure control and property optimization [11,17].

In order to overcome the disadvantages of coarse eutectic silicon, research has focused on methods including modification treatment, intermediate alloy, and short-term heat treatment [18,19,20]. However, these methods have shortcomings, mainly as follows: (1) the content of refining agents is difficult to control, and high content can easily form impurities; (2) the metamorphic process is prone to over-metamorphism, leading to grain coarsening, and; (3) short heat treatment time is difficult to control. It is worth noting that spark plasma sintering technology has the advantages of fast heating and low sintering temperature, which can effectively overcome these disadvantages. In addition, rapid heating using SPS can effectively inhibit the growth of eutectic silicon, thus obtaining fine dispersed eutectic silicon and improving the strength and toughness of the alloy.

In this study, SPS was used to fabricate AlSi10Mg alloy. Due to the rapid heating characteristics of SPS, grain refinement and dispersed distribution of eutectic silicon in nano-scale were expected to prevent the growth of eutectic silicon during SPS. The effect of sintering temperature on microstructure evolution and friction and wear properties was carefully analyzed. Electrochemical properties were also tested, and corresponding corrosion mechanisms were revealed.

## 2. Materials and Methods

Pre-alloyed AlSi10Mg powder was used in the experiment, with particle sizes ranging from 15 to 53 μm. The powder was sourced from Anhui Hart 3D Technology Co., Ltd., Wuhu, China. The chemical composition is shown in Table 1. The AlSi10Mg alloy was synthesized using SPS (LABOX-650F, SINTERLAND, Nagaoka, Japan). A K-type thermocouple was used to measure the temperature in the furnace. AlSi10Mg powder was poured into graphite molds, and graphite paper was used for mold releasing. Figure 1 shows the relationship between temperature, pressure, and time during the sintering process of AlSi10Mg alloy. The sintering pressure, soaking time, and heating rate were maintained at 45 MPa, 4 min, and (12.5 °C/min, 25.0 °C/min, 37.5 °C/min, and 50.0 °C/min), respectively. The sintered composite material was cooled to room temperature in the furnace and depressurized at 50 °C. The compact size was Φ 30 × 15 mm. The metallographic samples and friction and wear samples were cut from sintered bulk materials using wire-electrode cutting. After standard heat treatment (solid solution: 540 °C × 6 h and water cooling, aging: 160 °C × 6 h and air cooling), the microstructure analysis, mechanical property test, and electrochemical property test were carried out. For metallographic observation (OM, IE500M), the samples were mechanically ground with 2000# sandpaper, then mechanically polished and ultrasonically cleaned. Then, the samples were chemically etched using a solution of 10 vol%HF + 10 vol%HNO_3_ + 80 vol% H_2_O. The microstructure was further observed using an SEM (HITACHI SU8010, Tokyo, Japan) equipped with an EDS probe. X-ray diffraction (Bruker D8 ADVANCE, Billerica, MA, USA) Aikesaipu Instrument Equipment Co., Ltd. in Changsha, China was used for phase identification. The scanning angle was 20~90° and the scanning speed was 5°/min using Cu K_α_ radiation.

Microhardness was tested on an MVS-1000IMT2 Vickers hardness tester (Xiangxing Instrument Co., Ltd., Dongguan, China) with a load of 20 N and holding time of 10 s. The microhardness test points were evenly distributed on the surface of the sample, with a distance between test points of more than 0.1 mm, making the measurement results accurate. The average value was taken after 10 measurements. Friction and wear tests were conducted on a Zhongke Kaihua HT-1000 (Lanzhou, China) using a Φ 4 mm Si_3_N_4_ ceramic ball. The test load was 10 N and maintained for 15 min. The friction radius was 3 mm and rotation speed was 600 r/min. The wear surface was analyzed using SEM to reveal the wear mechanism.

The electrochemical performance of the AlSi10Mg alloy after heat treatment was tested on a CHI604E electrochemical workstation. The electrochemical testing was carried out using a three-electrode system. The aluminum alloy acted as the working electrode and platinum acted as the cathode, while silver chloride was used for the reference electrode. The electrolyte was a 3.5 wt% sodium chloride solution, and the corrosion contact surface was a sample with a size of Φ 4 × 6 mm. (The center of the sintered sample is along the axial line). The bottom was wrapped in copper foil and sealed with epoxy resin. The bottom of the cylindrical sample was exposed for electrolytic reaction. Before the experiment, the surface roughness of the sample after grinding (2000# sandpaper) and polishing was lower than Ra0.04, and it was cleaned with alcohol ultrasound. Then, the sample was placed in the electrolyte for 1 h. At the beginning of the electrochemical experiment, the open circuit potential (EoCp) was continuously measured until the system stabilized, followed by AC impedance testing and potentiodynamic polarization testing. The AC impedance test frequency was 10^−1^~10^5^ Hz, and the sine wave amplitude was determined to be 10 mV. A scanning rate range of 10~1000 mV/min and a scanning potential range of −1.5~0.5 V were applied during the dynamic potential polarization test. Zview 3.0 and CView 3.0 software were used to analyze the impedance spectrum and electric polarization curve, and SEM was used to observe the corrosion morphology.

## 3. Results

### 3.1. Powder Characterization

Figure 2 shows an SEM image of AlSi10Mg pre-alloyed powder.

The results indicated that the powder was mainly spherical, and irregular powder was also visible. The irregular powder formed because the incompletely solidified droplets knocked into each other during gas atomization. The statistical results indicate that the average diameter of the pre-alloyed powder was 21.50 μm.

### 3.2. Initial Microstructure

Figure 3 shows the XRD patterns of AlSi10Mg alloy sintered at different temperatures.

AlSi10Mg alloy is mainly composed of Al (JCPDS: NO.01-1176) and Si (JCPDS: NO.01-0787). In addition, the nominal content of Mg is 1%, and the specific content of Mg is less than 1%. Due to the low magnesium content, it is difficult to detect the phase of magnesium in the XRD spectra.

Figure 4 shows the microstructures of AlSi10Mg alloy sintered at different temperatures.

The sintered compact was dense without obvious macroscopic defects such as pores. The previous particle boundaries disappeared as the sintering temperature increased. The AlSi10Mg alloy was composed of α-Al and block precipitations, where α-Al was the matrix and the block precipitations were eutectic silicon phase. In addition, it was obvious that the block eutectic silicon phase was refined with the increase in sintering temperature. This was analyzed in detail using SEM.

Figure 5 shows SEM images of AlSi10Mg alloy sintered at different temperatures.

It can be observed that block precipitations in micro-scale and nano-scale were dispersed in the Al matrix. The micro-scale precipitations were determined to be eutectic silicon based on the EDS results. The nano-scale precipitations were not identified through EDS. The nano-scale precipitations were the Mg_2_Si particles in the Al-Si-Mg alloys according to the references [21,22]. The statistics results of grain size and eutectic silicon size are shown in Figure 5e. The chemical composition (at%) of EDS analysis in Figure 5b is shown in Table 2. The grain size at first decreased with the increase in sintering temperature, and then increased. The minimum grain size was about 9 μm for the sintering temperature of 525 °C. Moreover, the eutectic silicon size was refined to 2 μm. As the temperature rose to 550 °C, both grain and eutectic silicon obviously grew. The previous particle boundaries were obvious when the alloy was sintered at 475 °C, resulting in a large grain size as well as eutectic silicon. High temperature contributed to the diffusion and previous particle boundaries of eutectic silicon, which became invisible as the sintering temperature increased to 500 °C. As a result, grain refinement was achieved and the redissolution of eutectic silicon became pronounced. Then, a fine eutectic silicon formed. With the increase in sintering temperature, the grain growth accelerated, resulting in an increase in grain size. It can be concluded that sintering temperature significantly affects the grain growth and development of precipitates in the alloy, and AlSi10Mg alloy with fine grains and homogeneously precipitated eutectic silicon can be obtained by applying an appropriate sintering temperature.

### 3.3. Friction Performance

Figure 6 shows the friction coefficients of AlSi10Mg alloys sintered at different temperatures.

At the initial stage of friction, the friction coefficient was high because of the large friction resistance between the friction pair. The wear debris was compacted and the friction coefficient tended to be stable as the wear progressed [23,24]. In addition, with the increase in sintering temperature, the average friction coefficient first decreased and then increased. The average friction coefficient reached a low of 0.29 for the alloy sintered at 525 °C, which was a decrease of 22% compared to that of the alloy sintered at 475 °C. The main reason was that the grain size was smaller and the precipitates were uniformly dispersed in the matrix, which improved the wear resistance [25]. The average friction coefficients and microhardnesses of AlSi10Mg alloys sintered at different temperatures are shown in Figure 6b. The microhardness presented a tendency to first increase and then decrease. This is opposite to the changes in grain size and eutectic silicon size. It can be concluded that wear resistance is affected by microhardness, grain size, and the precipitations.

In order to analyze the wear mechanism, SEM was used to analyze the worn surface, as shown in Figure 7.

Peeling and shedding were observed on the worn surface of AlSi10Mg alloy sintered at 475 °C, indicating poor wear resistance (the average friction coefficient was 0.37). These results were consistent with the friction performance. After wear, a large number of furrows and adhesions appeared on the surface along the wear direction, indicating that the wear mechanism of AlSi10Mg alloy at room temperature was mainly adhesive wear [26]. The formation of furrows was attributed to the hard Si_3_N_4_ balls penetrating the soft aluminum matrix during the wear process, resulting in plastic deformation, as shown in Figure 7b,c. The adhesion arose because the Si_3_N_4_ balls bonded with the metal, and the Si_3_N_4_ balls gradually rolled during the sliding process.

The friction and wear debris were analyzed using SEM and EDS, as shown in Figure 8.

The surface after friction and wear showed deep grooves, indicating that the sliding wear presented typical ploughing characteristics. The EDS results are shown in Table 3.

The debris was composed of N, O, Mg, Al and Si. The N was derived from the hard Si_3_N_4_ balls penetrating the aluminum matrix. The N was added to the debris during the subsequent wear process. The O came from the atmosphere, and the oxide was formed during friction and wear. It can also be seen that the size of the debris for the AlSi10Mg alloy sintered at 525 °C was rather small, which revealed excellent wear resistance.

### 3.4. Electrochemical Performance

Table 4 shows the corrosion parameters of the potentiodynamic polarization experiment in a 3.5 wt% NaCl solution.

From the table, it can be seen that different sintering temperatures had a significant impact on the corrosion performance of AlSi10Mg alloy. Sintering at lower temperatures resulted in high corrosion current. As the sintering temperature increased, the corrosion potential moved in the positive direction and the corrosion current decreased. This indicated that high sintering temperature can improve the corrosion performance of the alloy. Figure 9a shows the potentiodynamic polarization curves.

The corrosion potential E_corr_ changed from −0.65V to −0.62V as the sintering temperature increased from 475 °C to 550 °C, indicating an increase in corrosion tendency. The corrosion current changed from 6.23 × 10^−6^ a·cm^−2^ to 4.51 × 10^−6^ a·cm^−2^. In Figure 9b, the Nyquist plot shows that all AlSi10Mg alloys were composed of a single capacitive arc. The diameter of the capacitor ring first increased and then decreased as the sintering temperature increased. Generally speaking, a larger diameter of the capacitor ring denoted a better corrosion resistance. The ring diameter of the AlSi10Mg alloy sintered at 500 °C was higher than that of the other three alloys. Bode and Bode-phase diagrams are shown in Figure 9c. Moreover, the impedance modulus |Z| of AlSi10Mg alloy sintered at 500 °C was higher than that of the other three alloys. In addition, the maximum phase angle was also higher than the other three alloys. This was consistent with the results of the potentiodynamic polarization experiment, indicating that the corrosion resistance of AlSi10Mg alloy sintered at 500 °C is better than those of such alloys sintered at other temperatures. To elucidate the effect of sintering temperature on the corrosion behavior of AlSi10Mg alloy, we studied the equivalent circuit, as shown in Figure 9d. R_s_, R_ct_, and CPE1 refer to electrolyte resistance, transfer charge resistance, and electrode surface double-layer capacitance, respectively. Table 4 lists the fitting results based on equivalent circuits (fitting error < 5%). The higher R_ct_ value indicates that it is more difficult for a charge to transfer. The decreasing trend of R_ct_ values for the four samples was: 500 °C > 525 °C > 550 °C > 475 °C.

The surface morphology of AlSi10Mg alloy after electrochemical corrosion is shown in Figure 10.

At low magnification, cracks and corrosion pits were observed on the corrosion surfaces of AlSi10Mg alloys sintered at different temperatures. Comparing the number of corrosion pits, the surface damage of the AlSi10Mg alloy sintered at 500 °C was relatively shallow. In addition, bright white cracked corrosion products were observed on the corrosion surface, which was due to poor connectivity with the surrounding corrosion products. The corrosion of aluminum alloys included pitting corrosion, galvanic corrosion and intergranular corrosion [27,28]. The formation of pits was observed on the corroded surface, which was attributed to the high penetrability of Cl^−^, and the formation of pits easily destroyed the oxide film. In addition, the research showed that the aluminum alloy suffered mainly pitting corrosion at the initial stage, and the pits then connected and expanded. Intergranular corrosion occurred due to the potential difference of precipitated phase (Si particles), grain boundary, and grain, which manifested as network corrosion. The aging precipitated Mg_2_Si had little effect on corrosion properties [28]. At high magnification, network-like corrosion and cracks wrapping around Si particles were observed, which was caused by potential difference. Using EDS, diffusion layers were found around Si particles of different sizes, which confirmed that the matrix around Si particles was corroded.

The sintering temperature determines the grain size, precipitation morphology, and size of AlSi10Mg alloy, indirectly affecting its corrosion performance. The statistical results show that as the sintering temperature increases, the grain size increases (Figure 5f), while the content of eutectic silicon significantly decreases. During low-temperature sintering, oxides and impurities aggregate at the original boundary, leading to potential difference and galvanic corrosion. Meanwhile, due to poor bonding between powders, corrosion through boundary channels intensifies and corrosion resistance significantly decreases. During high-temperature sintering, the original particle boundaries disappear, and the size and morphology of precipitates are the main factors. Generally speaking, the smaller the grain size, the higher the grain boundary (GB) content. Grain boundaries are related to grain distribution and phase structure, and they have an impact on optimal corrosion performance. In addition, the difference in chemical composition between grain boundaries and the matrix leads to a difference in electrochemical potential [29]. Fu et al. [30] improved the intergranular corrosion performance of TWIP steel from the perspective of GB, believing that the small grain size and complex distribution of ∑3GBs can improve corrosion performance. Yang et al. [31] studied the corrosion behavior of ultrafine eutectic Al-12Si. The results indicate that large-scale silicon can reduce the stability of the oxide film and exacerbate corrosion. Ultrafine silicon leads to the formation of a large number of micro couples, which is beneficial for the early formation of oxide films and protects the matrix alloy. The grain size and silicon particle size of the 500 °C alloy were higher than those of the 525 °C alloy, corresponding to the best corrosion resistance. However, the volume fraction of silicon particles also influences corrosion resistance. The volume fraction of silicon particles in the alloy sintered at 525 °C was lower than that of 500 °C, which is detrimental to corrosion resistance. Thus, the results indicate that AlSi10Mg alloy sintered at 500 °C shows the best corrosion resistance.

## 4. Conclusions

In this study, the effect of sintering temperature on the microstructural evolution, friction and wear properties, and electrochemical performance of AlSi10Mg alloy was systematically studied. The main conclusions are summarized as follows:(1)Sintering temperature had an important influence on the grain size, eutectic silicon size, and previous particle boundaries of AlSi10Mg alloys. The optimum sintering temperature was 525 °C, its resulting grain size was 9 μm, and the eutectic silicon size was 2 μm, which was uniformly dispersed in the matrix.(2)The AlSi10Mg alloy sintered at 525 °C exhibited excellent friction and wear properties. Its friction coefficient was 0.29, which was decreased by 22% compared to AlSi10Mg alloy sintered at 475 °C. The friction and wear mechanism of AlSi10Mg alloy was adhesive wear, and grain refinement and homogeneously precipitated micro-scale eutectic silicon can effectively improve wear resistance.(3)The AlSi10Mg alloy sintered at 500 °C exhibited excellent electrochemical performance based on the I_corr_ and E_corr_. Corrosion mainly occurred around eutectic silicon as well as the corrosion cracks due to the potential difference between eutectic silicon and the matrix.

## Figures and Tables

**Figure 1 materials-16-07394-f001:**
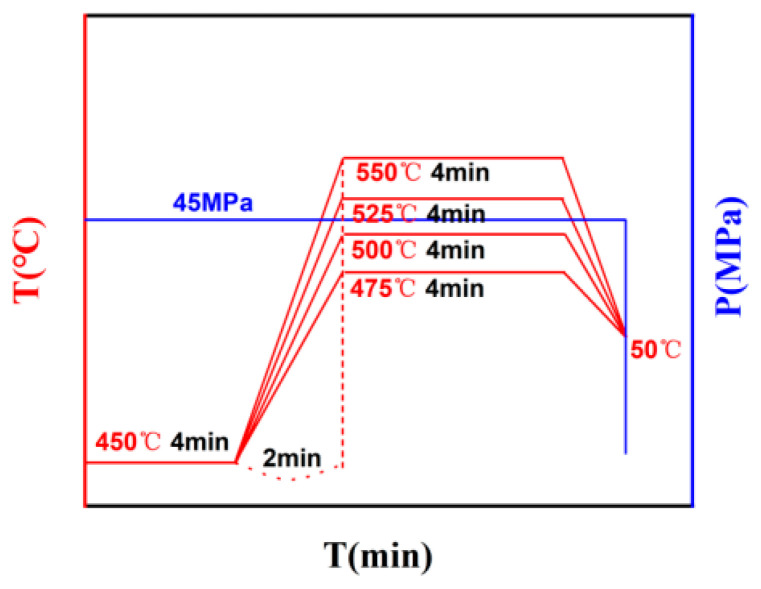
Relationship between temperature, pressure, and time during the sintering process of AlSi10Mg alloy.

**Figure 2 materials-16-07394-f002:**
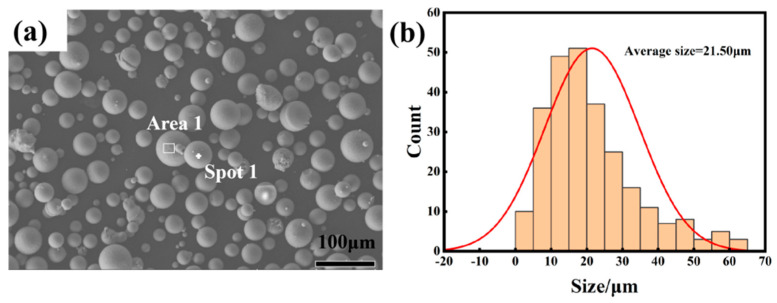
SEM images of AlSi10Mg pre-alloyed powder: (**a**) SEM image of AlSi10Mg pre-alloy powder; (**b**) Particle size statistics of AlSi10Mg pre-alloy powder.

**Figure 3 materials-16-07394-f003:**
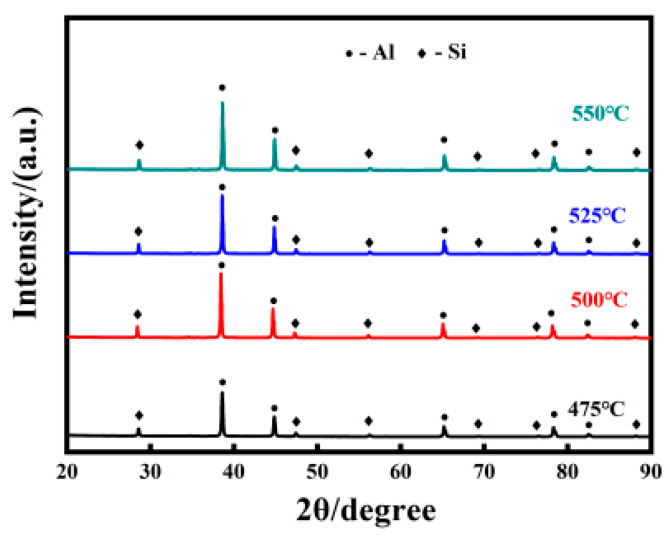
XRD patterns of AlSi10Mg alloy sintered at different temperatures.

**Figure 4 materials-16-07394-f004:**
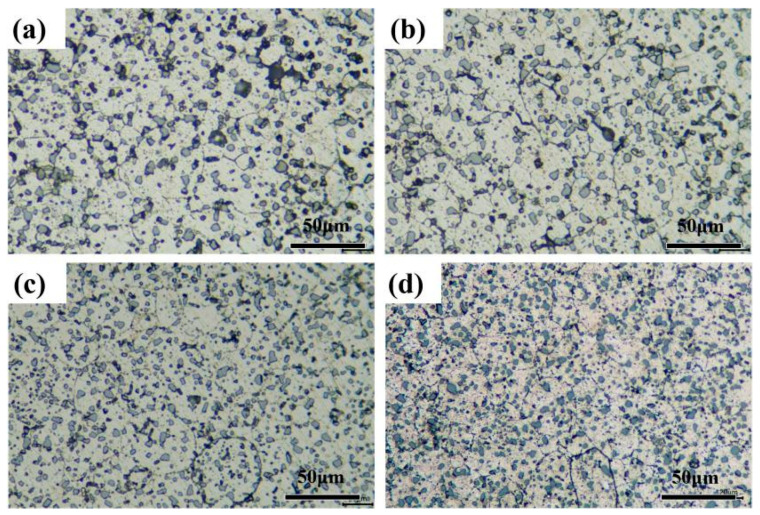
The microstructures (OM images) of the AlSi10Mg alloy sintered at different temperatures: (**a**) 475 °C; (**b**) 500 °C; (**c**) 525 °C; (**d**) 550 °C.

**Figure 5 materials-16-07394-f005:**
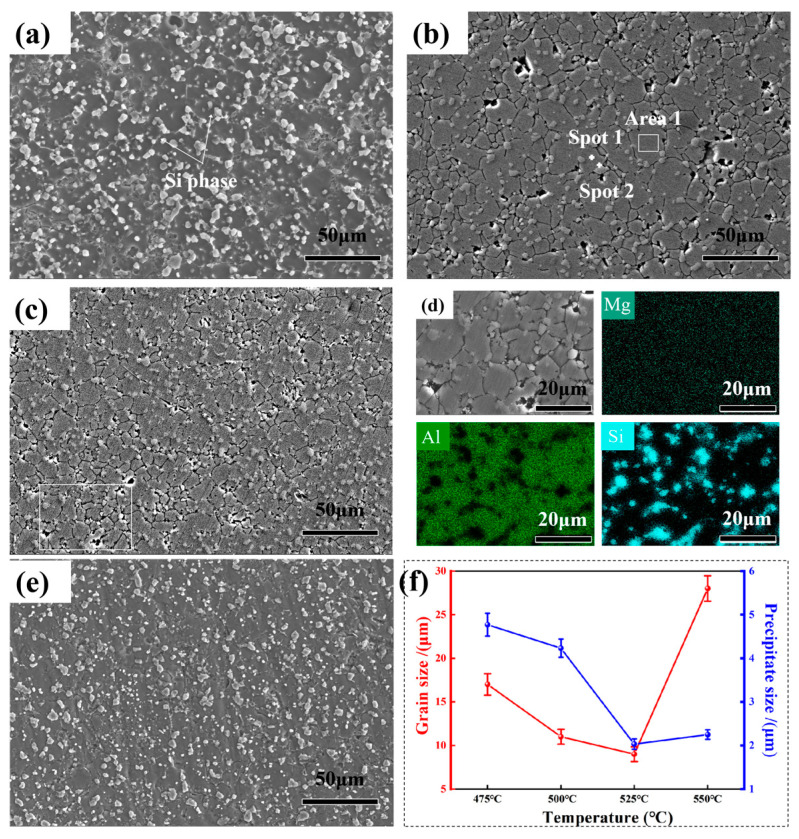
The microstructures (SEM images) of AlSi10Mg alloy with different sintering temperatures: (**a**) 475 °C; (**b**) 500 °C; (**c**) 525 °C; (**d**) enlarged morphology of (**c**) and mapping results; (**e**) 550 °C; (**f**) statistics of grain and precipitated phase size.

**Figure 6 materials-16-07394-f006:**
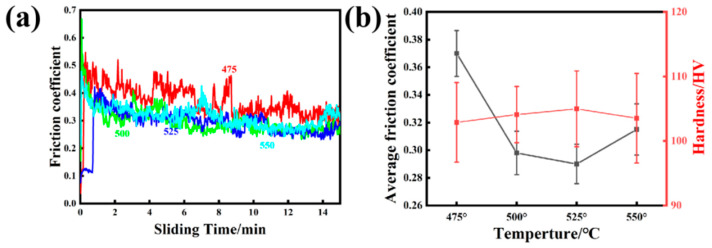
Room-temperature friction and wear of AlSi10Mg alloy sintered at different temperatures: (**a**) friction coefficient; (**b**) average friction coefficient and microhardness.

**Figure 7 materials-16-07394-f007:**
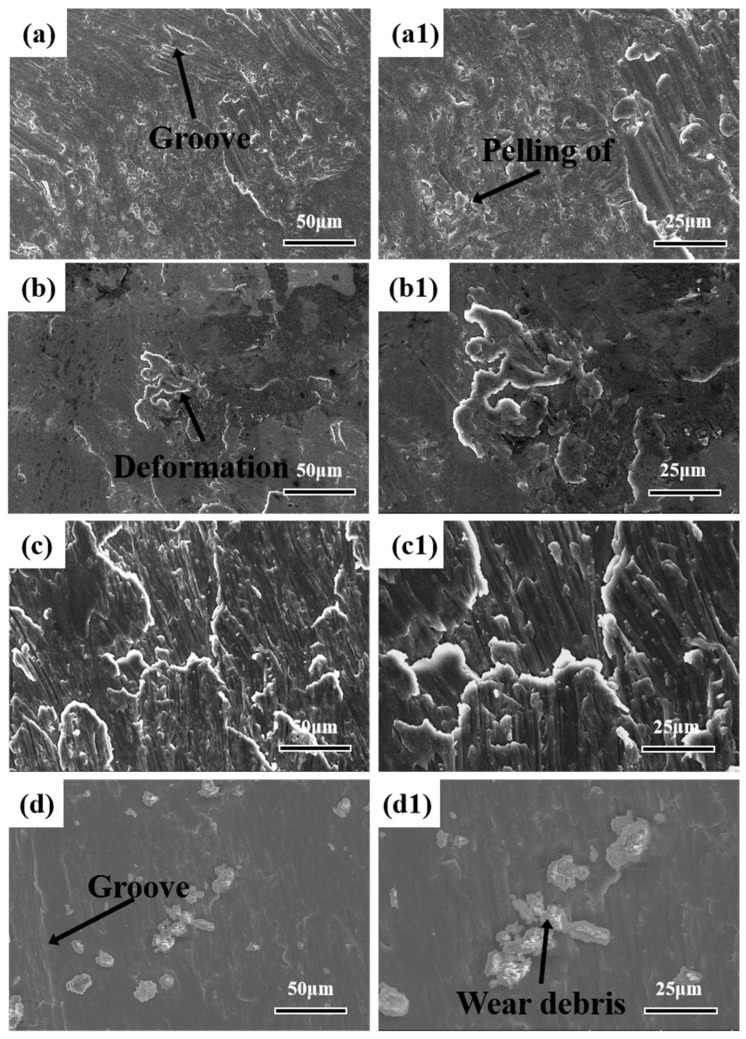
Wear surface of AlSi10Mg alloy sintered at different temperatures: (**a**,**a1**) 475 °C; (**b**,**b1**) 500 °C; (**c**,**c1**) 525 °C; (**d**,**d1**) 550 °C.

**Figure 8 materials-16-07394-f008:**
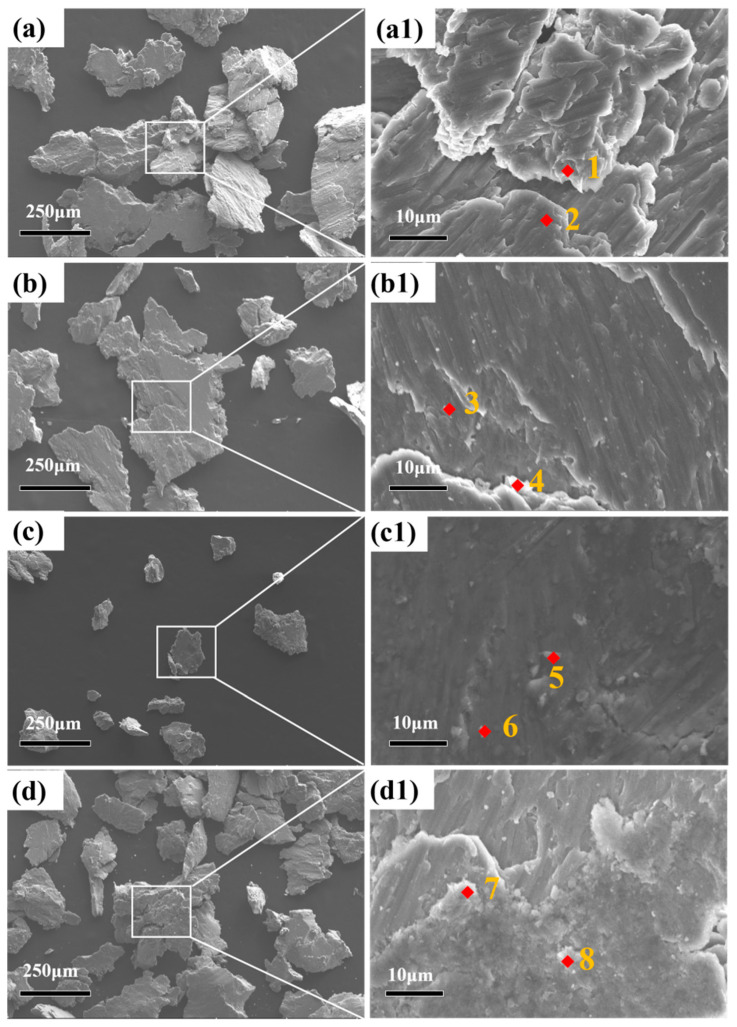
Friction and wear debris of AlSi10Mg alloys sintered at different temperatures: (**a**,**a1**) 475 °C; (**b**,**b1**) 500 °C; (**c**,**c1**) 525 °C; (**d**,**d1**) 550 °C; (Spot 1~8) Corresponding EDS spectral positions.

**Figure 9 materials-16-07394-f009:**
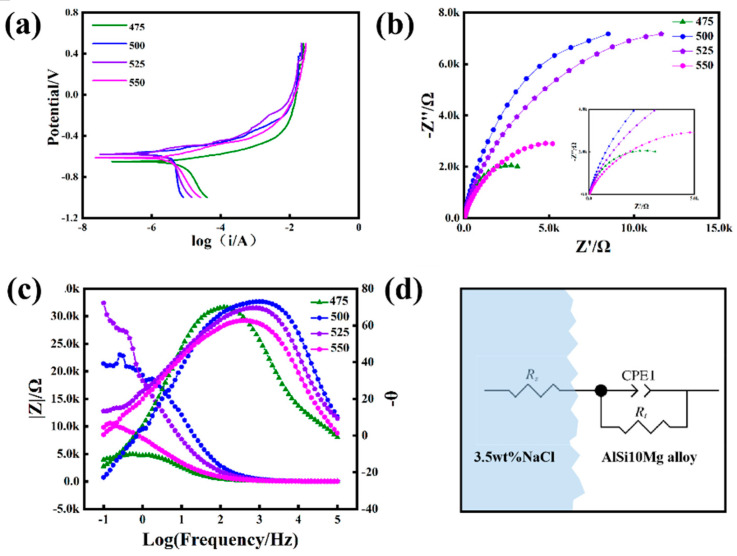
Polarization curves of AlSi10Mg alloys sintered at different temperatures: (**a**) potentiodynamic polarization curve; (**b**) Nyquist diagram; (**c**) Bode diagram; (**d**) circuit model.

**Figure 10 materials-16-07394-f010:**
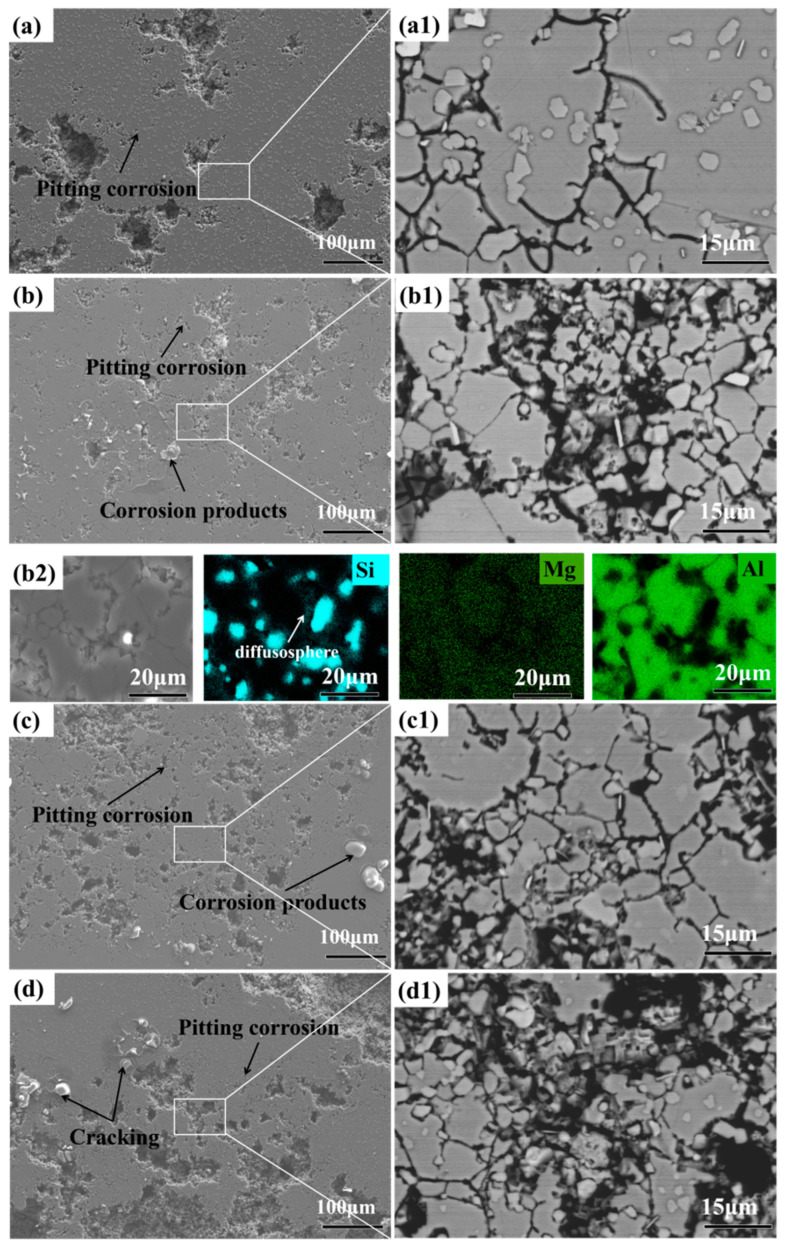
Electrochemical corrosion surface and EDS analysis of AlSi10Mg alloys sintered at different temperatures: (**a**,**a1**) 475 °C; (**b**,**b1**) 500 °C; (**b2**) corresponding EDS mapping results; (**c**,**c1**) 525 °C; (**d**,**d1**) 550 °C.

**Table 1 materials-16-07394-t001:** Chemical composition of AlSi10Mg powder standard and SEM corresponding to Figure 2a.

AlSi10Mg	Al	Si	Mg	Fe	Cu	Mn	Ti	Zn
Standard	Margin	9~11	0.2~0.45	≤0.55	≤0.05	≤0.45	≤0.15	≤0.10
Area 1	85.07	12.74	0.97	0.38	0.39	0.17	0.11	0.16
Spot 1	85.92	11.50	1.10	0.54	0.56	0.16	0.09	0.12

**Table 2 materials-16-07394-t002:** Chemical composition (at%) of EDS analysis in Figure 5b.

at%	Al	Si	Mg	Fe	Cu	Mn	Ti	Zn
Area 1	95.21	3.38	1.06	0.07	0.12	0.05	0.04	0.07
Spot 1	96.37	2.15	1.04	0.09	0.10	0.10	0.06	0.08
Spot 2	9.40	90.11	0.11	0.06	0.15	0.08	0.02	0.07

**Table 3 materials-16-07394-t003:** Chemical composition of different positions on the abrasive surface (at.%).

Positions	N	O	Mg	Al	Si
Point 1	0.33	0.45	0.79	95.93	2.51
Point 2	0.06	1.98	0.84	72.66	24.47
Point 3	5.73	34.24	0.01	47.53	12.49
Point 4	0.06	3.86	0.67	66.18	29.22
Point 5	1.90	26.56	0.20	64.00	7.34
Point 6	4.92	34.63	0.42	49.97	10.06
Point 7	0.72	8.82	0.47	83.53	6.45
Point 8	1.05	12.65	0.19	72.49	13.62

**Table 4 materials-16-07394-t004:** Electrochemical corrosion parameters and equivalent circuit fitting parameters of AlSi10Mg alloy at different sintering temperatures.

Sintering Temperature/°C	I_corr_/A·cm^−2^	E_corr_/V	R_s_/(Ω·cm^−2^)	CPE_dl_-T/10^−6^/(F·cm^−2^)	CPE_dl_-P	R_ct_/(Ω·cm^−2^)
475	6.23 × 10^−6^	−0.65	30.30	8.48	0.86	5148
500	1.33 × 10^−6^	−0.57	26.59	1.37	0.86	18,431
525	1.87 × 10^−6^	−0.58	28.14	3.16	0.81	16,712
550	4.51 × 10^−6^	−0.62	24.77	8.61	0.76	7630

## Data Availability

Data are contained within the article.

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
