# Peer review of "Exploring Microstructure, Wear Resistance, and Electrochemical Properties of AlSi10Mg Alloy Fabricated Using Spark Plasma Sintering"

_materials, 2023, doi:10.3390/ma16237394_

Round 1
Reviewer 1 Report
Comments and Suggestions for Authors
Dear Authors, I have got a few remarks/questions regarding text of the article.
1. Why the abstract does not include information about performer heat treatment? The title of the article deals with preparation of the alloy basing on the SPS (does performed heat treatment is the standard procedure in this case? – I have some doubts about this).
2. What is connection of the information written in the Introduction chapter with Your research (verse 46 to 57)? Taking into account the investigated alloy, there are number of publications connected with such alloy. You do not introduce any additional particles to the alloy, You only make use a standard powder (what is its chemical composition?)
3. „After standard heat treatment (solid solution: 540 72 °C × 6 h and water cooling, aging: 160 °C × 6 h and air cooling.) - verse 72” –on which basis such parameters were considered as the standard?
4. „The average value was taken after 10 measurements (verse 82)” – how the measurement points are distributed on the surface of the sample?
5. „… surface was a sample with size of Φ 4 × 6 mm (verse 92)” – similar question – from which area these samples were cut-off.
6. Figure 3. Is this material after the heat treatment? How did the structure look like after the SPS process SPS?
6. „The nano-scale precipitations were deduced to be Mg2Si phase according the references in the Al-Si-Mg system (verse 146) – please, justify this statement.
7. „… which was decreased by 21.89% com (verse 167)” – do the Authors are aware of what values we are talking in case of the hundredth parts of percent of change? Please round such results to a level of measuring accuracy of the device.
8. „The microhardness presented a tendency of increase firstly and then decrease. This was consensus with the change of grain size and eutectic silicon size (verse 171-172)” – trivial sentence having none reference in the analysed drawing, where changes of the parameters are at the level are at the level of the measurement errors/distribution of the measurement. Slight changes in the hardness are tried to associate with considerable (comparing to changes of the hardness) changes of the average friction coefficient.
9. Figure 6 – and what surface of the initial material look like?
10. Figure 7 – for what purpose the lines were introduced from corners of the marked area in the figures a-d?
11. Table 1 (verse 196) lacking information about Point 8.
12. Figure 9 (which area of the figures a-d was magnified?
13. „The grain size and precipitated phase optimization will be conducive to the improvement of corrosion performance (verse 272)” – on which basis such conclusion? Obtained results of the research work do not fully confirm this conclusion. The Authors did not analyse precipitated phases and their effect on the properties). Please explain this question.
Author Response
Responses to the comments by the editor and reviewers
Dear Nicoleta Dragomir,
Thank you very much for dealing with this Manuscript ID materials-2694215 carefully. Also, we thank the reviewer for his/her serious and pertinent comments, and it is no doubt that reviewer is really an expert on our research topic. We have carefully read these comments and thought them over. The responses are listed as follows. Besides, the corresponded revisions are highlighted in yellow in the resubmitted revised manuscript.
- Responses to comments from Reviewer #1
Question 1: Why the abstract does not include information about performer heat treatment? The title of the article deals with preparation of the alloy basing on the SPS (does performed heat treatment is the standard procedure in this case? – I have some doubts about this).
Response: Thanks for the comments. The heat treatment information has been added in the abstract. Al-Mg-Si ternary alloy is commonly used for casting alloy, and corresponding heat treatment (solution temperature: 525~550℃, aging temperature: 150~200℃) is to improve its mechanical properties. The alloying element and processing method influence the heat treatment parameters. This article studies the effect of sintering temperature on microstructure and properties of heat treated alloys. Then, the heat treatment parameters are determined as solid solution: 540 °C × 6 h and water cooling, aging: 160 °C × 6 h and air cooling according to the actual condition.
Question 2: What is connection of the information written in the Introduction chapter with Your research (verse 46 to 57)? Taking into account the investigated alloy, there are number of publications connected with such alloy. You do not introduce any additional particles to the alloy, You only make use a standard powder (what is its chemical composition?)
Response: Thanks for the comments. “According to Jia et al. [11], adding Sr and Y elements can improve the spheroidization and grain refinement of eutectic silicon.After adding 0.2% Y aluminum silicon alloy for heat treatment, the tensile strength and elongation increased by 14.3% and 8.3%, respectively, to 344MPa and 6.3%.Yin et al. [12] reported that Sn element addition can refine and modify eutectic silicon particles. Heat treatment is also used to improve mechanical properties and spheroidized Si particles [13]. However, the relationship between the electrochemical performance and grain size of Al-Si-Mg alloy, as well as eutectic silicon, is complex. In addition, for cast aluminum alloys, it is also necessary to consider the influence of secondary dendrites [14]. Kim et al. [15] studied the effect of ultrasonic melting treatment on the corrosion properties of A356 alloy, which is attributed to the uniform dispersion of refined al grains, SDAS and eutectic Si. Du et al. [16] found that the morphology and distribution of silicon in Al-Si alloys are directly related to their corrosion resistance. Specifically, when the morphology and distribution of silicon particles are more uniform, their corrosion resistance is improved. In addition, a more uniform and fine distribution of silicon particles will be beneficial for improving the corrosion resistance of Al-Si alloys.Therefore, controlling the size and morphology of eutectic silicon in Al-Si-Mg alloy is crucial for microstructure control and property optimization [11,17]”. The introduction has been rewritten. In this study, SPS technology was used to prepare AlSi10Mg alloy. Due to the rapid heating characteristics of SPS, the growth of eutectic silicon was avoided. Thereby finely dispersed eutectic silicon in nanoscale was obtained and improved the strength and toughness of the alloy. The influence of sintering temperature on microstructure evolution and friction wear performance was studied, and electrochemical performance was tested, revealing the corresponding corrosion mechanism. This study provided us with a deeper understanding of Al-Si-Mg alloys, and new ideas for microstructure controlling. The raw materials used in this study were purchased from Anhui Hart 3D Technology Co., Ltd., and this was added in the experimental section. The chemical composition was shown in Tab 1.
Tab. 1. Chemical composition of AlSi10Mg powder
AlSi10Mg |
Al |
Si |
Mg |
Fe |
Cu |
Mn |
Ti |
Zn |
Standard |
Margin |
9~11 |
0.2~0.45 |
≤0.55 |
≤0.05 |
≤0.45 |
≤0.15 |
≤0.10 |
Question 3: “After standard heat treatment (solid solution: 540 °C × 6 h and water cooling, aging: 160 °C × 6 h and air cooling.) - verse ” –on which basis such parameters were considered as the standard?
Response: Thanks for the comments.The inaccurate description has been revised as “After heat treatment (solid solution: 540 °C × 6 h and water cooling, aging: 160 °C × 6 h and air cooling)”.
Question 4:“The average value was taken after 10 measurements (verse 82)” – how the measurement points are distributed on the surface of the sample?
Response: Thanks for the comments.“The microhardness test points are evenly distributed on the surface of the sample, with a distance between each test point more than 0.1mm, making the measurement results accurate”.
Question 5: “surface was a sample with size of Φ 4 × 6 mm (verse 92)” – similar question – from which area these samples were cut-off.
Response: Thanks for the comments.The size of the sintered sample is Φ30 ×15 mm, and Φ4 × 6mm sample is taken from the center of the sintered sample along the axial direction, this has been added in the experimental section.
Question 6: Figure 3. Is this material after the heat treatment? How did the structure look like after the SPS process SPS?
Response: Thanks for the comments.The initial microstructure of AlSi10Mg alloy after SPS was shown below. The original boundary of the powder can be observed in the sintered compact, and a large amount of precipitated phases distributed in grains.
The initial microstructure of AlSi10Mg alloy after SPS
Question 7:“The nano-scale precipitations were deduced to be Mg2Si phase according the references in the Al-Si-Mg system (verse 146) – please, justify this statement.
Response: Thanks for the comments.The statement has been revised as “The nano-scale precipitations were the Mg2Si particles in the Al-Si-Mg alloys according the references [21,22]”.After solution and aging heat treatment, Mg2Si particles can precipitate in Al-Si-Mg alloy, which contributes to its mechanical properties. Similar results have been cited and the nano precipitates are identified to be Mg2Si phase.
References
[21]Ragab. KhA, Samuel. AM, Al-Ahmari. AMA, Samuel. FH, Doty. HW, Effect of Multi-Temperature Aging on the Characterization of Aluminum. Based Castings Heat Treated Using Fluidized Bed Technique.Metals and Materials International, 2013,19(04),783-802. https://doi.org/ 10.1007/s12540-013-4019-1.
[22]Emamy. M, Malekan. M, Pourmonshi. AH, Tavighi. K, The influence of heat treatment on the structure and tensile properties of thin-section A356 aluminum alloy casts refined by Ti, B and Zr. Journal of Materials Research,2017,32(18), 3540–3547. https://doi.org/ 10.1557/jmr.2017.193 .
Question 8: “… which was decreased by 21.89% com (verse 167)” – do the Authors are aware of what values we are talking in case of the hundredth parts of percent of change? Please round such results to a level of measuring accuracy of the device.
Response: Thanks for the comments.Corresponding results have been revised to a level of measuring accuracy of the device.
Question 9: “The microhardness presented a tendency of increase firstly and then decrease. This was consensus with the change of grain size and eutectic silicon size (verse 171-172)” – trivial sentence having none reference in the analysed drawing, where changes of the parameters are at the level are at the level of the measurement errors/distribution of the measurement. Slight changes in the hardness are tried to associate with considerable (comparing to changes of the hardness) changes of the average friction coefficient.
Response: Thanks for the comments.By analyzing the changes of microhardness, grain size, eutectic silicon size, and average friction coefficient, it is possible to better understand the relationship between the microstructure and properties. The results indicate that there is a correlation between the average friction coefficient and microhardness. In addition, it is also necessary to consider the possible errors and uncertainties that exists during experimental process.
Question 10:Figure 6 – and what surface of the initial material look like?
Response:Thanks for the comments.“the surface roughness of the sample after grinding (2000 # sandpaper) and polishing was lower than Ra0.04”.
Question 11:Figure 7 – for what purpose the lines were introduced from corners of the marked area in the figures a-d?
Response: Thanks for the comments.For further observation, locally enlarged regions was selected for high magnification observation.
Figure 8. Friction and wear debris of AlSi10Mg alloys sintered at different temperatures: (a-a1) 475 °C; (b-b1) 500 °C; (c-c1) 525 °C; (d-d1) 550 °C.
Question 12: Table 1 (verse 196) lacking information about Point 8.
Response: Thanks for the comments.The lacking information about Point 8 has been added in the revised manuscript, and we have carefully reviewed and revised the manuscript.
Table 3. Chemical composition of different positions on the abrasive surface (at.%).
Positions |
N |
O |
Mg |
Al |
Si |
Point 1 |
0.33 |
0.45 |
0.79 |
95.93 |
2.51 |
Point 2 |
0.06 |
1.98 |
0.84 |
72.66 |
24.47 |
Point 3 |
5.73 |
34.24 |
0.01 |
47.53 |
12.49 |
Point 4 |
0.06 |
3.86 |
0.67 |
66.18 |
29.22 |
Point 5 |
1.90 |
26.56 |
0.20 |
64.00 |
7.34 |
Point 6 |
4.92 |
34.63 |
0.42 |
49.97 |
10.06 |
Point 7 |
0.72 |
8.82 |
0.47 |
83.53 |
6.45 |
Point 8 |
1.05 |
12.65 |
0.19 |
72.49 |
13.62 |
Question 13: Figure 9 (which area of the figures a-d was magnified?
Response: Thanks for the comments.We have made modifications to the enlarged area.
Figure 10. Electrochemical corrosion surface and EDS analysis of AlSi10Mg alloys sintered at different temperatures:(a-a1) 475 °C; (b-b1) 500 °C; (b2) corresponding EDS mapping results; (c-c1) 525 °C;(d-d1) 550 °C.
Question 14: “The grain size and precipitated phase optimization will be conducive to the improvement of corrosion performance (verse 272)” – on which basis such conclusion? Obtained results of the research work do not fully confirm this conclusion. The Authors did not analyse precipitated phases and their effect on the properties). Please explain this question.
Response: This conclusion is mainly based on the following discussion:“Generally speaking, the smaller the grain size, the higher the grain boundary (GB) content. Grain boundaries are related to grain distribution and phase structure, and have an impact on optimal corrosion performance. In addition, the difference in chemical composition between grain boundaries and the matrix leads to a difference in electrochemical potential [28]. Fu et al. [30] improved the intergranular corrosion performance of TWIP steel from the perspective of GB, believing that the small grain size and complex distribution of ∑3GBs can improve the corrosion performance. Yang et al. [31] studied the corrosion behavior of ultrafine eutectic Al-12Si. The results indicate that large-scale silicon can reduce the stability of the oxide film and exacerbate corrosion. Ultrafine silicon leads to the formation of a large number of micro couples, which is beneficial for the early formation of oxide films and protects the matrix alloy. The grain size and silicon particle size of the 500 ℃ alloy are higher than those of the 525 ℃ alloy, corresponding to the best corrosion resistance. However, the volume fraction of silicon particle also influences the corrosion resistance. The volume fraction of silicon particle in the alloy sintered at 525 ℃ was lower than that of 500℃, then it is detrimental to corrosion resistance. Then, the results indicate that AlSi10Mg alloy sintered at 500 ℃ shows the best corrosion resistance”.
We hope that our responses could meet the requirements of the reviewer and you. And if there are other questions about our revised manuscript, please don’t hesitate to contact us.
Thank you and the reviewers very much.
Kind regards,
Corresponding author:
Tengfei Ma
Key Laboratory of Air-driven Equipment Technology of Zhejiang Province, Quzhou University, Quzhou 324000, China
Tel.: +86-571-8026716.
E-mail: matengfeihit@163.com
Corresponding author:
Xiaoying Jiang
Key Laboratory of Air-driven Equipment Technology of Zhejiang Province, Quzhou University, Quzhou 324000, China
E-mail address: qz_jxy@163.com

Reviewer 2 Report
Comments and Suggestions for Authors
The reviewed manuscript entitled “Microstructure and properties of AlSi10Mg alloy prepared by spark plasma sintering” delves into the optimization of AlSi10Mg alloy microstructure through Spark Plasma Sintering and its influence on wear resistance and corrosion mechanisms. The article demonstrates a commendable scientific and technical quality. However, in order to enhance the manuscript's overall readability and clarity, several significant concerns must be addressed before proceeding with further processing:
1- Title: It would be advantageous to specify which specific properties of the AlSi10Mg alloy will be investigated. For example: "Exploring Microstructure, Wear Resistance and Electrochemical properties of AlSi10Mg Alloy Fabricated by Spark Plasma Sintering"
2- The abstract requires some quantitative brief results. The abstract is a mini version of manuscript that proceeds. So, include introduction, methodology, results and concluding remarks in a precise but effective manner.
3- The motivation for the study and the research gap are not clear enough. Please demonstrate in the introduction of the paper, the novelty of this research in relation to other thematically similar research papers.
4- Avoid lumped references; a short comment should be included for each reference or two references in the same subject.
5- The composition of the alloy is mentioned to contain 10% Mg. However, it is crucial to specify the silicon content in the alloy. An EDS analysis is required as an evidence of the alloy's chemical composition and verify its constituents.
6- Figure 1, SEM: The two figures marked as "a" and "b" do not appear to be closely matching. Please provide the original SEM image with the original scale bar, magnification and other printed information.
7- Section 2. Materials and Methods: “The pre-alloyed AlSi10Mg powder was used in the experiment and the particle size was 15~53 μm.” Further details regarding the source of this pre-alloyed powder should be included. It is essential to specify whether the alloy was prepared by the authors themselves or purchased from a specific manufacturer. Additionally, the inclusion of Energy Dispersive X-ray Spectroscopy (EDS) mapping is recommended to validate the purity and homogeneity of the powder mixture, which is a crucial aspect of ensuring the reliability of the experimental materials.
8- L 70: It would be valuable to provide information regarding the heating rate used to attain the various sintering temperatures. This detail is essential for understanding the thermal processing conditions and their potential influence on the resulting microstructure and properties of the AlSi10Mg alloy.
9- L84: “The friction radius was 3 mm and rotation speed was 600 r/min.” It is important to address why a small friction radius of 3 mm was chosen, especially considering that the fabricated sample has a diameter of 30 mm. Using a larger friction radius could significantly improve the investigation by providing a wider frictional track, which may yield more comprehensive and representative data.
10- Figure 2, XRD: Please identify all peaks in the XRD patterns, providing the unique powder diffraction file (PDF) or JCPDS Card Number for each element/compound/phase in the pattern.
11- Why Mg did not detected in the XRD analysis? The absence of magnesium (Mg) in the XRD analysis, despite its high percentage of 10% in the alloy, raises a noteworthy concern. It's essential to address and provide an explanation for this absence, as it may impact the interpretation of the alloy's crystalline structure and composition.
12- In Figure 4, specifically in the subfigure "f," there is a noticeable significant increase in grain size at 550°C, whereas the optical microscope image does not appear to reflect this difference. Please provide clarification regarding this discrepancy.
13- Discussion is lack of scientific explanation for the obtained results. Authors should attribute the results achieved to a clear scientific reason.
14- Please elaborate on the correlation between the fluctuations in corrosion potential (Ecorr) and corrosion current (Icorr) with the changing sintering temperature? It would be insightful to understand how these electrochemical parameters are linked to the microstructural variations observed at different sintering temperatures. Specifically, how does the microstructure, which includes factors such as grain size and eutectic silicon distribution, influence the corrosion resistance behavior of the AlSi10Mg alloy at various temperatures?
15- The English language used in the paper is to be revised and improved before the subsequent manuscript submission. Please, read the text carefully before the next submission of the paper.
Author Response
Responses to the comments by the editor and reviewers
Dear Nicoleta Dragomir,
Thank you very much for dealing with this Manuscript ID materials-2694215 carefully. Also, we thank the reviewer for his/her serious and pertinent comments, and it is no doubt that reviewer is really an expert on our research topic. We have carefully read these comments and thought them over. The responses are listed as follows. Besides, the corresponded revisions are highlighted in yellow in the resubmitted revised manuscript.
- Responses to comments from Reviewer #2
Question 1: Title: It would be advantageous to specify which specific properties of the AlSi10Mg alloy will be investigated. For example: "Exploring Microstructure, Wear Resistance and Electrochemical properties of AlSi10Mg Alloy Fabricated by Spark Plasma Sintering"
Response: Thanks for the comments. The title has been revised as“Exploring Microstructure, Wear Resistance and Electrochemical properties of AlSi10Mg Alloy Fabricated by Spark Plasma Sintering”.
Question 2: - The abstract requires some quantitative brief results. The abstract is a mini version of manuscript that proceeds. So, include introduction, methodology, results and concluding remarks in a precise but effective manner.
Response: Thanks for the comments.The abstract has been rewritten.“Al-Si-Mg alloy has excellent casting performance due to its high silicon content, but the coarse eutectic silicon phase can lead to a decrease in its mechanical properties. AlSi10Mg alloy was prepared by spark plasma sintering method, and it was found that the sintering temperature has a significant impact on the grain size, eutectic silicon size, wear and corrosion properties after heat treatment. At a sintering temperature of 525° C, it exhibits the best wear performance with an average friction coefficient of 0.29. This is attributed to the uniform precipitation of fine eutectic silicon phases, significantly improving wear resistance, and determining that the wear mechanism of AlSi10Mg alloy at room temperature is adhesive wear. The electrochemical performance of AlSi10Mg sintered at 500 ° C is the best, with Icorr and Ecorr being 1.33×10-6A·cm-2 and -0.57V, respectively. This is attributed to the refinement of grain size and eutectic silicon size, as well as the appropriate Si volume fraction. Therefore, optimizing the sintering temperature can effectively improve the performance of the alloy”.
Question 3: - The motivation for the study and the research gap are not clear enough. Please demonstrate in the introduction of the paper, the novelty of this research in relation to other thematically similar research papers.
Response: Thanks for the comments.We have modified the introduction section.“According to Jia et al. [11], adding Sr and Y elements can improve the spheroidization and grain refinement of eutectic silicon.After adding 0.2% Y aluminum silicon alloy for heat treatment, the tensile strength and elongation increased by 14.3% and 8.3%, respectively, to 344MPa and 6.3%.Yin et al. [12] reported that Sn element addition can refine and modify eutectic silicon particles. Heat treatment is also used to improve mechanical properties and spheroidized Si particles [13]. However, the relationship between the electrochemical performance and grain size of Al-Si-Mg alloy, as well as eutectic silicon, is complex. In addition, for cast aluminum alloys, it is also necessary to consider the influence of secondary dendrites [14]. Kim et al. [15] studied the effect of ultrasonic melting treatment on the corrosion properties of A356 alloy, which is attributed to the uniform dispersion of refined al grains, SDAS and eutectic Si.Du et al. [16] found that the morphology and distribution of silicon in Al-Si alloys are directly related to their corrosion resistance. Specifically, when the morphology and distribution of silicon particles are more uniform, their corrosion resistance is improved. In addition, a more uniform and fine distribution of silicon particles will be beneficial for improving the corrosion resistance of Al-Si alloys. Therefore, controlling the size and morphology of eutectic silicon in Al-Si-Mg alloy is crucial for microstructure control and mechanical property optimization [11,17].
In order to overcome the disadvantage of coarse eutectic silicon, research has focused on these methods, including modification treatment, intermediate alloy, and short-term heat treatment [18-20]. However, these methods have shortcomings, mainly as follows: 1) the content of refining agents is difficult to control, and high content can easily form impurities; 2) The metamorphic process is prone to over metamorphism, leading to grain coarsening; 3) Short time heat treatment time is difficult to control. It is worth noting that spark plasma sintering technology has the advantages of fast heating and low sintering temperature, which can effectively overcome this disadvantage. In addition, rapid heating by SPS can effectively inhibit the growth of eutectic silicon, thus obtaining fine dispersed eutectic silicon and improving the strength and toughness of the alloy”.
Question 4: Avoid lumped references; a short comment should be included for each reference or two references in the same subject.
Response: Thanks for the comments. The lumped references were adjusted.
Question 5: The composition of the alloy is mentioned to contain 10% Mg. However, it is crucial to specify the silicon content in the alloy. An EDS analysis is required as an evidence of the alloy's chemical composition and verify its constituents.
Response: Thanks for the comments. The written of AlSi10Mg is confused, and composition of the alloy is Al-10Si-1Mg.“Figure 5 shows the microstructure of AlSi10Mg alloy at different sintering temperatures (SEM images)”and“Table 1 shows the chemical composition (at%) of EDS analysis in Figure 5 (b)”.
Figure 5. The microstructures (SEM images) of the AlSi10Mg alloy with different sintering temperatures: (a) 475 °C; (b) 500 °C; (c) 525 °C;(d) enlarged morphology of fig c and mapping results; (e) 550 °C; (f) statistics of grain and precipitated phase size.
Table 1. Chemical composition (at%) of EDS analysis in Figure 5 (b)
at% |
Al |
Si |
Mg |
Fe |
Cu |
Mn |
Ti |
Zn |
Area 1 |
95.21 |
3.38 |
1.06 |
0.07 |
0.12 |
0.05 |
0.04 |
0.07 |
Spot 1 |
96.37 |
2.15 |
1.04 |
0.09 |
0.10 |
0.10 |
0.06 |
0.08 |
Spot 2 |
9.40 |
90.11 |
0.11 |
0.06 |
0.15 |
0.08 |
0.02 |
0.07 |
Question 6: Figure 1, SEM: The two figures marked as "a" and "b" do not appear to be closely matching. Please provide the original SEM image with the original scale bar, magnification and other printed information.
Response: Thanks for the comments. The original image of AlSi10Mg powder is shown below.
Original image of AlSi10Mg powder
Question 7: Section 2. Materials and Methods: “The pre-alloyed AlSi10Mg powder was used in the experiment and the particle size was 15~53 μm.” Further details regarding the source of this pre-alloyed powder should be included. It is essential to specify whether the alloy was prepared by the authors themselves or purchased from a specific manufacturer. Additionally, the inclusion of Energy Dispersive X-ray Spectroscopy (EDS) mapping is recommended to validate the purity and homogeneity of the powder mixture, which is a crucial aspect of ensuring the reliability of the experimental materials.
Response: Thanks for the comments. “Pre-alloyed AlSi10Mg powder was used in the experiment, with particle sizes ranging from 15 to 53 μ m. Originating from Anhui Hart 3D Technology Co., Ltd. The chemical composition is shown in Table 1 ”and“The statistical results indicate that the average diameter of the pre-alloyed powder is 21.50 μm”.
Table 1. Chemical composition of AlSi10Mg powder standard and SEM corresponding to Fig 1 (a)
AlSi10Mg |
Al |
Si |
Mg |
Fe |
Cu |
Mn |
Ti |
Zn |
Standard |
Margin |
9~11 |
0.2~0.45 |
≤0.55 |
≤0.05 |
≤0.45 |
≤0.15 |
≤0.10 |
Area 1 |
85.07 |
12.74 |
0.97 |
0.38 |
0.39 |
0.17 |
0.11 |
0.16 |
Spot 1 |
85.92 |
11.50 |
1.10 |
0.54 |
0.56 |
0.16 |
0.09 |
0.12 |
Question 8: - L 70: It would be valuable to provide information regarding the heating rate used to attain the various sintering temperatures. This detail is essential for understanding the thermal processing conditions and their potential influence on the resulting microstructure and properties of the AlSi10Mg alloy.
Response: Thanks for the comments.“Figure 1 shows the relationship between temperature, pressure, and time during the sintering process of AlSi10Mg alloy. The sintering pressure, soaking time, and heating rate were maintained at 45MPa, 4 minutes, and (12.5 ℃/min, 25.0 ℃/min, 37.5 ℃/min, and 50.0 ℃/min), respectively. The sintered composite material is cooled to room temperature in the furnace and depressurized at 50 ℃”.
Figure 1. shows the relationship between temperature, pressure, and time during the sintering process of AlSi10Mg alloy.
Question 9: - L84: “The friction radius was 3 mm and rotation speed was 600 r/min.” It is important to address why a small friction radius of 3 mm was chosen, especially considering that the fabricated sample has a diameter of 30 mm. Using a larger friction radius could significantly improve the investigation by providing a wider frictional track, which may yield more comprehensive and representative data.
Response: Thanks for the comments.The sintered compact was cut into smaller samples for heat treatment and electrochemical experiment. Therefore, a small friction radius of 3 mm was chosen.
Question 10: - Figure 2, XRD: Please identify all peaks in the XRD patterns, providing the unique powder diffraction file (PDF) or JCPDS Card Number for each element/compound/phase in the pattern.
Response: Thanks for the comments.Figure 3 showed XRD patterns of AlSi10Mg alloy sintered at different sintering temperatures.“AlSi10Mg alloy is mainly composed of Al (JCPDS: NO.01-1176) and Si (JCPDS: NO.01-0787). In addition, the nominal content of Mg is 1%, and the specific content of Mg is less than 1%. Due to the low content of magnesium, it is difficult to detect the phase of magnesium in XRD spectra”.
Figure 3. XRD patterns of AlSi10Mg alloy sintered at different temperatures.
Question 11: - Why Mg did not detected in the XRD analysis? The absence of magnesium (Mg) in the XRD analysis, despite its high percentage of 10% in the alloy, raises a noteworthy concern. It's essential to address and provide an explanation for this absence, as it may impact the interpretation of the alloy's crystalline structure and composition.
Response: Thanks for the comments.“The nominal content of Mg is 1%, and the specific content of Mg was less than 1%. Due to its low content, it is difficult to detect the phase of Mg element in XRD spectra”.
Table 1. Chemical composition of AlSi10Mg powder
AlSi10Mg |
Al |
Si |
Mg |
Fe |
Cu |
Mn |
Ti |
Zn |
Standard |
Margin |
9~11 |
0.2~0.45 |
≤0.55 |
≤0.05 |
≤0.45 |
≤0.15 |
≤0.10 |
Question 12: - In Figure 4, specifically in the subfigure "f," there is a noticeable significant increase in grain size at 550°C, whereas the optical microscope image does not appear to reflect this difference. Please provide clarification regarding this discrepancy.
Response: Thanks for the comments.We have changed the image to be clearer.
Figure 4. The microstructures (OM images) of the AlSi10Mg alloy sintered at different temperatures: (a) 475 ℃; (b) 500 ℃; (c) 525 ℃;(d) 550 ℃.
Question 13: - Discussion is lack of scientific explanation for the obtained results. Authors should attribute the results achieved to a clear scientific reason.
Response: Thanks for the comments.The discussion was rewritten.
Question 14: - Please elaborate on the correlation between the fluctuations in corrosion potential (Ecorr) and corrosion current (Icorr) with the changing sintering temperature? It would be insightful to understand how these electrochemical parameters are linked to the microstructural variations observed at different sintering temperatures. Specifically, how does the microstructure, which includes factors such as grain size and eutectic silicon distribution, influence the corrosion resistance behavior of the AlSi10Mg alloy at various temperatures?
Response: Thanks for the comments. We have made modifications in the article.There is a certain correlation between the corrosion potential (Ecorr) and corrosion current (Icorr) of AlSi10Mg alloy and the changes in sintering temperature.“Generally speaking, the smaller the grain size, the higher the grain boundary (GB) content. Grain boundaries are related to grain distribution and phase structure, and have an impact on optimal corrosion performance. In addition, the difference in chemical composition between grain boundaries and the matrix leads to a difference in electrochemical potential [28]. Fu et al. [29] improved the intergranular corrosion performance of TWIP steel from the perspective of GB, believing that the small grain size and complex distribution of ∑3GBs can improve the corrosion performance. Yang et al. [30] studied the corrosion behavior of ultrafine eutectic Al-12Si. The results indicate that large-scale silicon can reduce the stability of the oxide film and exacerbate corrosion. Ultrafine silicon leads to the formation of a large number of micro couples, which is beneficial for the early formation of oxide films and protects the matrix alloy. The grain size and silicon particle size of the 500 ℃ alloy are higher than those of the 525 ℃ alloy, but we did not consider the effects of factors such as grain boundary angle, morphology, and volume fraction. The results indicate that AlSi10Mg alloy sintered at 500 ℃ has the best corrosion resistance, and the optimization of grain size and precipitation phase will help improve the corrosion performance”.
Question 15: - The English language used in the paper is to be revised and improved before the subsequent manuscript submission. Please, read the text carefully before the next submission of the paper.
Response: Thanks for the comments.The manuscript has been checked carefully, and we have asked a native English speaker to revise and edit the paper before it is resubmitted to the journal this time. The revised parts are highlighted in revised manuscript, we hope that can meet the reviewer’s and magazine’s standard.
These have been added in revised manuscript.
We hope that our responses could meet the requirements of the reviewer and you. And if there are other questions about our revised manuscript, please don’t hesitate to contact us.
Thank you and the reviewers very much.
Kind regards,
Corresponding author:
Tengfei Ma
Key Laboratory of Air-driven Equipment Technology of Zhejiang Province, Quzhou University, Quzhou 324000, China
Tel.: +86-571-8026716.
E-mail: matengfeihit@163.com
Corresponding author:
Xiaoying Jiang
Key Laboratory of Air-driven Equipment Technology of Zhejiang Province, Quzhou University, Quzhou 324000, China
E-mail address: qz_jxy@163.com

Round 2
Reviewer 2 Report
Comments and Suggestions for Authors
The revision is satisfactory and the authors have provided amendments to all the suggested queries. Therefore, I recommend this work for publication in Materials journal.